# Fitness Costs of the Glutathione *S*-Transferase Epsilon 2 (L119F-GSTe2) Mediated Metabolic Resistance to Insecticides in the Major African Malaria Vector *Anopheles Funestus*

**DOI:** 10.3390/genes9120645

**Published:** 2018-12-19

**Authors:** Magellan Tchouakui, Jacob M. Riveron, Doumani Djonabaye, Williams Tchapga, Helen Irving, Patrice Soh Takam, Flobert Njiokou, Charles S. Wondji

**Affiliations:** 1LSTM Research Unit at the Centre for Research in Infectious Diseases (CRID), P.O. Box 13591 Yaoundé, Cameroon; jacob.riveron_miranda@syngenta.com (J.M.R.); doumani_dd@yahoo.com (D.D.); wills.tchapga@gmail.com (W.T.); njiokouf@yahoo.com (F.N.); 2Parasitology and Ecology Laboratory, Department of Animal Biology and Physiology, Faculty of Science, University of Yaoundé 1, P.O. Box 812 Yaoundé, Cameroon; 3Department of Vector Biology, Liverpool School of Tropical Medicine, Pembroke Place, Liverpool L35QA, UK; helen.irving@lstmed.ac.uk; 4Department of Biochemistry, Faculty of Science, University of Yaoundé 1, P.O. Box 812 Yaoundé, Cameroon; 5Department of Mathematics, Faculty of Science, University of Yaoundé 1, P.O. Box 812 Yaoundé, Cameroon; b_calvo2002@yahoo.fr

**Keywords:** malaria, vector control, *Anopheles funestus*, metabolic resistance, fitness cost, glutathione *S*-transferase, L119F-GSTE2

## Abstract

Metabolic resistance to insecticides threatens malaria control. However, little is known about its fitness cost in field populations of malaria vectors, thus limiting the design of suitable resistance management strategies. Here, we assessed the association between the glutathione *S*-transferase GSTe2-mediated metabolic resistance and life-traits of natural populations of *Anopheles funestus*. A total of 1200 indoor resting blood-fed female *An. funestus* (F_0_) were collected in Mibellon, Cameroon (2016/2017), and allowed to lay eggs individually. Genotyping of F1 mosquitoes for the L119F-GSTE2 mutation revealed that L/L119-homozygote susceptible (SS) mosquitoes significantly laid more eggs than heterozygotes L119F-RS (odds ratio (OR) = 2.06; *p* < 0.0001) and homozygote resistant 119F/F-RR (OR = 2.93; *p* < 0.0001). L/L119-SS susceptible mosquitoes also showed the higher ability for oviposition than 119F/F-RR resistant (OR = 2.68; *p* = 0.0002) indicating a reduced fecundity in resistant mosquitoes. Furthermore, L119F-RS larvae developed faster (nine days) than L119F-RR and L119F-SS (11 days) (X^2^ = 11.052; degree of freedom (df) = 4; *p* = 0.02) suggesting a heterozygote advantage effect for larval development. Interestingly, L/L119-SS developed faster than 119F/F-RR (OR = 5.3; *p* < 0.0001) revealing an increased developmental time in resistant mosquitoes. However, genotyping and sequencing revealed that L119F-RR mosquitoes exhibited a higher adult longevity compared to RS (OR > 2.2; *p* < 0.05) and SS (OR > 2.1; *p* < 0.05) with an increased frequency of GSTe2-resistant haplotypes in mosquitoes of D30 after adult emergence. Additionally, comparison of the expression of *GSTe2* revealed a significantly increased expression from D1-D30 after emergence of adults (Anova test (F) = 8; df= 3; *p* = 0.008). The negative association between GSTe2 and some life traits of *An. funestus* could facilitate new resistance management strategies. However, the increased longevity of GSTe2-resistant mosquitoes suggests that an increase in resistance could exacerbate malaria transmission.

## 1. Introduction

Malaria remains one of the main causes of morbidity and mortality in Sub-Saharan Africa, predominantly in children under 5 years old and pregnant women [1]. Insecticide-based control interventions using pyrethroids and dichlorodiphenyltrichloroethane (DDT), notably through long lasting insecticide nets (LLINs) and indoor residual spraying (IRS), are key components of malaria control in Africa [1]. This strategy was recently revealed to have contributed to a decrease of more than 70% of malaria cases in the past decade [2]. Unfortunately, malaria vectors such as *Anopheles funestus* are increasingly developing resistance to the main insecticide classes particularly against pyrethroids, the only class recommended for bed net impregnation since they are safe and fast acting [3]. To ensure the continued effectiveness of insecticide-based interventions, it is crucial to design and implement suitable insecticide resistance management (IRM) strategies. However, designing such IRM strategies requires a good understanding of the fitness costs incurred by the development of resistance in the field populations. A fitness cost means that an individual possessing the resistance allele would lack some other advantages or “qualities” such that only susceptible insects will have such qualities in the absence of insecticide selection pressure [4]. In fact, it was shown that mutations or genes conferring resistance, such as the resistance of malaria vectors to insecticides, are usually associated with fitness costs that disrupt normal physiological functions of the vectors. For example, resistant vectors may have lower mating success [4,5], lower fecundity and fertility, higher developmental time and reduced longevity [6,7,8,9]. The presence of such fitness costs that can impact the spread and persistence of resistance alleles in the vector populations is a pre-requisite for the implementation of most insecticide resistance management strategies (IRMS) including rotation of insecticides [1]. Some progress has been made to study the fitness costs incurred by target-site resistance mechanisms [4,5,6,7,8], however, little is known about the fitness cost incurred by metabolic resistance [10], a mechanism that has been acknowledged to be more likely to lead to control failure [11]. This lack of information on the fitness cost of metabolic resistance is mainly caused by the absence of molecular markers to easily track the effect of this resistance in mosquitoes. In contrast, for target-site resistance, the first DNA-based diagnostic tools were available more than 20 years ago particularly for the knockdown resistance (*kdr*) [12]. However, recent efforts have contributed to diagnostic tools with the identification of the first DNA-based metabolic resistance marker in the major malaria vector *An. funestus* where a leucine to phenylalanine amino acid change at codon 119 in the glutathione *S*-transferase epsilon 2 (L119F-GSTe2) was demonstrated to confer resistance to pyrethroid/DDT [13]. The L119F-GSTe2 diagnostic assay provides an excellent tool to study the fitness costs of a metabolic-mediated resistance in natural populations of *An. funestus*. This mosquito species plays a major role in the transmission of malaria and is widely distributed across the continent [14]. The important role of *An. funestus* in malaria transmission is related to the high *Plasmodium falciparum* parasite infection rates (more than 5%) of this vector in many African countries including Cameroon, its wide distribution and its anthropophilic behavior [15,16]. Pyrethroid resistance has also been increasingly reported in *An. funestus* populations from different regions in Sub-Saharan Africa including in southern Africa (South Africa [17,18], Mozambique [19,20], Malawi [21,22]). It has also been reported in East Africa (Uganda and Kenya [23,24] and Tanzania [25]), Central (Cameroon [26,27,28]) and West Africa (Benin [29,30], Ghana [31,32], Senegal [33] and Nigeria [30]). From the field evidence, it is unclear how this increased report of resistance affects the life traits of the vectors and malaria control. Although entomological parameters suggest that resistance may lead to a failure to reduce the number of mosquitoes and the biting rate, there is little evidence of failure to control malaria [10]. In this study the fitness cost of insecticide resistance on natural populations of *An. funestus* was assessed by investigating several fitness components of life-history using field-collected mosquitoes from Cameroon where the *An. funestus* populations are both resistant to pyrethroids and DDT [28]. Fitness cost evaluated by comparing the life traits parameters between different genotypes of the L119F-GSTe2 marker revealed that L119F-GSTe2 mutation has a detrimental impact on some life-traits of *An. funestus* field mosquitoes including fecundity and development of larvae but in contrast increased the adult longevity.

## 2. Materials and Methods

### 2.1. Study Site and Sample Collection

Indoor resting female mosquitoes were collected between May 2016 and February 2017 in Mibellon (6°46′ N, 11°70′ E), a village in Cameroon located in the Adamawa Region; Mayo Banyo Division and Bankim sub-division. The main malaria vector in the area is *An. funestus* present throughout the year due to the presence of a lake, with minor contributions from *Anopheles gambiae* [28]. The main vector control approach used in the area is LLINs, and the villages benefited from the universal LLIN coverage campaigns in recent years (2015) before this study. These malaria vectors are resistant to pyrethroids and DDT [28]. Most of the houses are mainly made of mud and brick walls, with thatched or iron-sheet roofs. The communities rely mainly on subsistence farming, cultivating rice and maize but also small-scale fishing. The F_0_ females collected were kept in the insectary for at least four days and then left to oviposit using the forced-egg laying method as previously described [23]. All stages were reared according to the protocol previously described [23].

### 2.2. Life Traits Experiments

#### 2.2.1. Fecundity and Fertility

Fully gravid females collected were put individually in 1.5 mL Eppendorf tubes with damp filter paper to enable them to lay eggs. After oviposition, the number of eggs laid per female and the number of larvae were recorded. After assessing the normality of eggs distribution using a Shapiro–Wilk normality test, the impact of resistance on fecundity was assessed by comparing the median number of eggs laid by different genotypes using a Kruskal-Wallis non-parametric test. Odds ratio for oviposition between wild homozygote resistant mosquitoes (L119F-RR), homozygote susceptible (L119F-SS) and heterozygote mosquitoes (L119F-RS) was also assessed using a statistical significance calculation based on the Fisher’s exact probability test. The impact of resistance on fertility was assessed by comparing the hatch rate between different genotypes using a Chi-square (X^2^) test.

#### 2.2.2. Development of Larvae and Pupae

After recording the total number of larvae produced per female, all larvae from the different genotypes were pooled and reared in the same container, thus avoiding variation in environmental conditions. This experiment was performed in three replicates of 10 trays per replicate and all immature stages were reared in the standard insectary condition. Larval bowls were large enough to allow for a sufficient surface area to prevent overcrowding and competition for food. The number of larvae varied between 200 and 300 per tray and water was changed every two days in each tray to minimize the effect of pollution. Changes in the length of larval and pupal development times and mortality rates were equally assessed by genotyping a set of 150 larvae at different stages (L1, L2, L3 and L4). Genotype frequency were monitored in larvae, pupae and adults to assess the impact of L119F mutation on developmental time and mortality. The rates of pupae formation was evaluated by comparing the genotype and allele frequency from the starting of pupation (pupae D9), on the third day (pupae D11) and on the fifth day of pupation (pupae D13).

#### 2.2.3. Longevity of the Adult Mosquitoes

After adult emergence, mosquitoes were divided into three replicates. A set of about 40 mosquitoes was removed in each of the three replicates at different time points (day 1, 10, 20 and 30 after emergence). In average, 50 mosquitoes were used for genotyping whereas 3 pools of 10 mosquitoes each were used to assess the gene expression level of *GSTe2* at each time point. The lifespan of homozygous resistant adult mosquitoes was compared to that of susceptible and heterozygote mosquitoes by assessing the frequency of 119F resistant allele and expression levels of *GSTe2* (quantitative reverse transcription PCR; qRT-PCR) at different time points. In addition, the entire *GSTe2* gene of 882 bp was sequenced in 12 randomly selected mosquitoes for each time-point (D1, D10, D20 and D30 post-emergence) to assess the variation in haplotype diversity at the four-time points.

### 2.3. DNA Extraction and Species Identification

Genomic DNA (gDNA) was extracted from whole female mosquitoes (F_0_) and all larval and pupal stages using the LIVAK method [34]. All females used for oviposition were morphologically identified as belonging to the *An. funestus* group [14]. Molecular identification was achieved through a cocktail PCR described by Koekoemoer et al. (2002) in order to determine the species [35].

### 2.4. Detection of Plasmodium Parasite in F_0_ Field-Collected Mosquitoes

TaqMan assay protocol described by Bass et al. (2008) [36] was used to detect the presence of *Plasmodium* parasite in 200 *An. funestus* s.s field-collected females. This method detects the presence of *P. falciparum* (falcip+) and/or *P. ovale, P. vivax and P. malariae* (OVM+).

### 2.5. Genotyping of L119F-GSTe2 Mutation

To assess the role of metabolic-mediated resistance on the different life traits of resistant mosquitoes, the L119F-GSTe2 mutation, previously shown to confer DDT and permethrin resistance in *An. funestus* [13] was genotyped using a newly designed allele-specific PCR (AS-PCR) diagnostic assay. All the F_0_ field-collected mosquitoes oviposited and non- oviposited and all the mosquitoes from larval and pupal stages were genotyped. Two outer and two inner primers are needed for the AS-PCR. The inner primers were designed manually with mismatched nucleotides in the 3rd nucleotide from the 3′ end. PCR was carried out using 10 mM of each primer and 1 μL of gDNA as template in 15 μL reaction containing 10X Kapa Taq buffer A, 0.2 mM dNTPs, 1.5 mM MgCl_2_, 1 U Kapa Taq (Kapa Biosystems, Wilmington, MA, USA). The cycle parameters were: 1 cycle at 95 °C for 2 min; 30 cycles of 94 °C for 30 s, 58 °C for 30 s, 72 °C for 1 min and then a final extension at 72 °C for 10 min. PCR products were separated on 2% agarose gel by electrophoresis. The bands corresponding to different genotypes were interpreted as described by Tchouakui et al. (2018) [37].

### 2.6. Gene Expression Profile of GSTe2 and Longevity Adult of Adult Mosquitoes Using Quantitative Reverse Transcription Polymerase Chain Reaction

Total RNA from three biological replicates of D1, D10, D20 and D30 after the emergence of the adult was extracted using the Picopure RNA Isolation Kit (Life Technologies, Camarillo, CA, USA). The qRT-PCR assays were performed to assess the expression level of GSTe2 from D1 to D30; 1 mg of RNA from each of the three biological replicates made of pools of 10 mosquitoes at each time point, and FANG (full susceptible strain) was used as a template for cDNA synthesis using the superscript III (Invitrogen, Carlsbad, CA, USA) following the manufacturer’s guidelines. The qRT-PCR was carried out as previously described [38,39] with the relative expression level and fold-change (FC) of *GSTe2* in each time point relative to the susceptible strain calculated according to the 2^−ΔΔCT^ method [40] after normalization with the housekeeping genes ribosomal protein S7 (RSP7; AFUN007153) and actin 5C (AFUN006819).

### 2.7. Genetic Diversity of *GSTe2* Gene and Adult Longevity

The full-length of the *An. funestus GSTe2* gene (809 bp) was amplified from a total of 48 mosquitoes (12 for each time point). Two primers; Gste2F, 5′GGAATTCCATATGACCAAGCTAGT TCTGTACACGCT3′ and Gste2R, 5′TCTAGATCAAGCTTTAGCATTTTCCTCCTT3′ was used for gene amplification in 15 µL reaction containing 10 mM of each primer, 10X Kapa Taq buffer A, 0.2 mM dNTPs, 1.5 mM MgCl_2_, 1 U Kapa Taq (Kapa Biosystems). PCR conditions were 1 cycle at 95 °C for 5 min; 30 cycles of 94 °C for 30 s, 55 °C for 30 s, 72 °C for 1 min and then a final extension at 72 °C for 10 min. PCR products were firstly visualized on 1.5% agarose gel stained with Midori green stain (Nippon Genetics Europe, Dueren, Germany) and then purified using ExoSAP (New England Biolabs, Ipswich, UK) clean up protocol according to manufacturer recommendations and directly sequenced on both strands. Sequences were visualised and corrected using BioEdit v7.2.5 software [41] and aligned using ClustalW Multiple Alignment integrated with BioEdit [42]. Parameters of genetic diversity were assessed using DnaSP v5.10.01 software [43] and MEGA v7.0.21 software [44].

### 2.8. Data Analysis

All analyses were conducted using GraphPad Prism version 7.00 and R 3.3.2. for Windows.

## 3. Results

### 3.1. Field Collection and Species Identification

One thousand and two hundred blood-fed females were collected indoor in Mibellon, Cameroon. Results from PCR species identification performed on the F_0_ females morphologically identified as *An. funestus* group confirmed that they all belong to the major malaria vector, *An. funestus* s.s. species.

### 3.2. Infection of An. funestus by Plasmodium Parasite

Two-hundred field collected females were screened for *P. falciparum* (falcip+) and *P. ovale/P. vivax/P. malariae* (OVM+) using TaqMan (Applied Biosystems, Foster City, CA, USA) assay on whole mosquitoes. Twenty one percent (42/200) of mosquitoes were infected with *Plasmodium* parasites comprising 71% (30/42) infection by falcip+, 19.04% (8/42) by OVM+ and 9.52% (4/42) mixed infection by falcip+ and OVM+ using the whole mosquitoes. However, the sporozoite infection rate was 4.2% (5/120) with three falcip+, one OVM+ and one mix infection with falcip+/OVM+.

### 3.3. Genotyping of the L119F-GSTe2 in Field-Collected Mosquitoes

The L119F-GSTe2 was successfully genotyped in 260 oviposited females (F_0_) revealing a low frequency of the 119F resistant allele in the *An. funestus* s.s. population from Mibellon (24.8%). 6.1% (16/260) of the individuals were homozygote for the 119F resistant allele (119F/F-RR) whereas 37.3% (97/260) were heterozygote (L119F-RS), and 56.5% (147/260) were homozygote for the L119 susceptible allele (L/L119-SS).

### 3.4. Assessment of the Association between the L119F-GSTe2 Mutation and the Life Traits of An. funestus

#### 3.4.1. Association between L119F-GSTe2 and Fecundity/Fertility of Female Mosquitoes

In order to assess the role of the L119F mutation on the ability of wild *An. funestus* to lay eggs, we compared the frequency of the resistant allele between oviposited and non-oviposited females. This revealed a higher but not significant (X^2^ = 1.65; *p* = 0.19) frequency of 119F resistant (31.5%) in non-oviposited females compared to the oviposited females (24.8%) (Appendix A). Assessment of the odds ratio (OR) showed that the ability of L/L119-SS mosquitoes to lay eggs was higher compared to L119F-RS (OR = 2.06; confidence interval (CI) 95%: 1.45–2.92; *p* < 0.0001) and 119F/F-RR (OR = 2.93; CI 95%: 1.66–5.18; *p* < 0.0001) suggesting an association between the L119F mutation and reduced fecundity. Moreover, L119F-RS showed also a higher ability to lay eggs compared to L119F-RR (OR = 2.68; CI 95%: 1.51–4.77; *p* = 0.0002) (Table 1) suggesting an additional burden of the 119F allele on fecundity.

Furthermore, the mean number of eggs laid per female for 119F-RR was 65.8 (min = 12; max = 125. The mean was 95.7 with a clutch size ranging from 13 to 156 for L119F-RS while L119-SS laid a mean number of 93.5 eggs per female with a clutch size ranging from 2 to 162 (Figure 1a). Comparison of the mean number of eggs produced per genotype showed that 119F-RR mosquitoes produced a slightly, and significantly, lower number of eggs compared to L119F-RS and L119-SS (*p* = 0.003) (Figure 1c). Concerning the viability of eggs laid, the hatch rate was (65.8 ± 5.6) for 119F-RR, (66.0 ± 2.9) for L119F-RS and (62.9 ± 2.3) for L119F-SS mosquitoes (Figure 1b). The mean number of larvae was not different between genotypes (*p* = 0.18) as well as for the hatch rate (0.000 < X^2^ < 0.80; 0.79 < *p* < 0.98) (Figure 1c).

#### 3.4.2. Assessment of the Association between the L119F-GSTe2 Mutation and Larval Development

Egg-hatching occurred at 1–3 days (post-oviposition) and development time from the larvae to the pupae was 12.5 ± 4.5 days overall. Genotyping of 150 mosquitoes (50 per replicate) in each larval stage revealed that 4.9% ± 0.7% of mosquitoes were 119F-RR mosquitoes represented in L1, 5.7% ± 0.02% in L2, 6.3% ± 2.3% in L3 and 5.6% ± 1.07% in L4. L119F-RS mosquitos were 34.8% ± 2.5% of the mosquitoes in L1, 38.4% ± 1.2% in L2, 33.8% ± 4.6% in L3 and 35.6% ± 4.3% in L4 and L119-SS mosquitoes were 58.8% ± 6.5% of the mosquitoes in L1, 51.2% ± 5.9% in L2, 59.3% ± 4.9% in L3 and 56.4% ± 5.4% in L4. Comparison of genotype frequency from L1 to L4 larval stage showed no significant difference for 119F-RR (X^2^ < 0.27; *p* > 0.60), L119F-RS (X^2^ < 0.30; *p* > 0.57) and L119-SS mosquitoes (X^2^ < 1.21; *p* > 0.27) (Figure 2a,b). Furthermore, there was no significant change in the allele frequency (X^2^ < 0.81; *p* > 0.36) indicating that possessing the 119F resistant allele probably does not impact the larval development from L1 to L4 in this *An. funestus* population. Pupae were obtained from 9 days post-hatching (pupae D9) to 17 days (pupae D17) with most pupation (more than 75%) observed at 11 days post-hatching (pupae D11) (Figure 2d).

Assessment of the rate of pupae formation by comparing the frequency of the genotypes of the pupae obtained in D9, D11 and D13 showed that L119F-RS heterozygote mosquitoes developed significantly faster than homozygote resistant and homozygote susceptible mosquitoes (X^2^ = 11.052; degree of freedom (df) = 4; *p* = 0.02) indicating a possible heterozygote advantage (Appendix A; Figure 2c,d). Assessment of the OR for pupae formation further supported that L119F-RS developed significantly faster than L119F-RR (OR > 1.04; *p* < 0.42) and slightly faster than L119-SS although not significant (OR > 1.38; *p* < 0.08). However, L119-SS developed faster than 119F-RR (OR > 1.40; *p* < 0.0001) suggesting a potential fitness cost of the L119F-GSTe2 on the development of larvae (Table 2, Figure 2d).

#### 3.4.3. Assessment of the Association between L119F-GSTe2 Mutation and Adult Longevity

The lifespan of adult female mosquitoes F_1_ varied from 12 to 36 days for the three replicates. Comparison of the survival curve in term of adult mortality using a Log-rank (Mantel–Cox) test showed no difference between the three replicates (X^2^ = 0.2; *p* = 0.9) (Figure 3a).

Fifty live mosquitoes were genotyped at D1, D10, D20 and D30 after the emergence of adults to assess the association between the L119F mutation and adult longevity. Comparison of genotypes frequency showed a decrease proportion of L119F-SS homozygote susceptible mosquitoes from D1 to D30 (X^2^ = 21.2; *p* = 0.0017) (Figure 3b,d). Assessing the OR showed that mosquitoes with the 119F resistant allele lived longer compared to those with L119 susceptible allele (OR = 7.5; CI 95%: 1.04–21.3; *p* < 0.001), Table 3. This was supported by the variation in allele frequency after genotyping and sequencing (Figure 3e). In addition, mosquitoes with RR genotype had more chance to survive until D30 compared to RS (OR > 2.2; *p* < 0.05) and SS (OR > 2.1; *p* < 0.05) but no difference was observed between RS and SS. Evaluation of the expression level of *GSTe2* at the same time points showed also a significant level of expression of this gene in D30 (FC = 4.4 ± 2.7) than in D1 (FC = 2.5 ± 0.7), D10 (FC = 2.7 ± 0.8), D20 (FC = 2.8 ± 0.4) (F = 8; df = 3; *p* = 0.008) suggesting that mosquitoes expressing this gene live longer than those with lower expression (Figure 3c).

#### 3.4.4. Association between *GSTe2* Polymorphism and Longevity

##### Genetic Diversity of the *GSTe2*

A total of 809 bp fragments of the full length of *GSTe2* were successfully sequenced in 48 mosquitoes from Mibellon (12 mosquitoes at each time point) but only 44 sequences were successfully analysed (2*n* = 88). The genetic diversity parameters of the full fragment of *GSTe2* sequences are given in Table 4 according to the different time point. Overall, 12 polymorphic sites (11 in the coding and one in the non-coding regions) defining 33 haplotypes were detected. Mosquitoes of D30 after the emergence of the adults showed a lower number of the polymorphic site (9) with a reduced haplotype diversity (hd; 10 haplotypes hd: 0.84). The overall nucleotide diversity was 0.004 with an average number of differences between nucleotides estimated at 3.21 showing no significant differences between the sequences examined (*p* > 0.10). However, at D30 the L119F mutation was detected at very high frequency compared to D1 and D10 where the mutant allele was present at low frequency (54% compared to 29% in D1, 24 in D10 and 50 in D20) (X^2^ = 23.53 *p* < 0.0001), Figure 3d. This supports the assertion that the 119F resistant allele is associated with increased longevity in these field mosquitoes

#### 3.4.5. Distribution of Haplotypes and Phylogeny

Analysis of the haplotype network of the *GSTe2* gene based on L119F alleles and adult longevity showed that there are five major haplotypes with a frequency ≥5% (H1, H6, H8, H9 and H10) in this *An. funestus* field population (Figure 4a–c). Among the five major haplotypes, the ancestor haplotype (H6) belonged to the L119 susceptible allele and was shared between mosquitoes of D1, D20 and D30. The haplotype H1 with the highest frequency (14) belonged also to the susceptible allele and was common to D1, D10, and D30. H10 the second major haplotype was shared between mosquitoes of D1, D10, D20 and D30 and was specific to the 119F resistant allele. The H8 was specific to the L119 susceptible allele and the H9 was specific to the 119F resistant allele (Figure 4b,c). The analysis of the maximum likelihood phylogenetic tree between haplotypes identified did not reveal any haplotype clustering associated with a specific time point (Figure 4d). However, there was a trend of clustering according to the L119F mutation of the *GSTe2* gene (Figure 4e).

## 4. Discussion

Fitness costs incurred in the life-traits of resistant malaria mosquitoes through metabolic resistance have so far been difficult to establish due to lack of suitable molecular markers. Using the recent L119F-GSTe2 diagnostic tool for glutathione *S*-transferase metabolism of pyrethroids/DDT resistance in the resistant African malaria vector *An. funestus*, we showed in this study using mosquitoes collected in the same location that metabolic resistance could incur fitness costs in resistant mosquitoes but also provide a fitness advantage to resistant mosquitoes; although further work is needed to assess any possible effects associated with closely linked genes. This complex pattern offers a hope of managing such resistance if suitable resistance management strategies are implemented, but on the other hand it highlights the possible increase in malaria transmission risk as GSTe2-resistant adult mosquitoes live longer.

### 4.1. Association between L119F Resistance Marker and Adult Longevity

Longevity of adult vectors is a primary life trait for which a change due to fitness cost could impact the disease transmission risk as the vectors have to live sufficiently longer to be able to ingest the parasite, and harbor it while it develops until the infective stage [45]. Increased longevity was observed in this study in females possessing the 119F resistant allele. Such increased longevity is likely to increase the vectorial capacity of 119F-GSTe2 mosquitoes as the extrinsic incubation period of *Plasmodium* parasites is more likely to be completed and these females could take further blood meals with the infective sporozoite stage. However, such decreased mortality is not generally seen for other resistance markers in mosquitoes, such as the *kdr* in *Aedes aegypti* which instead was associated with decreased longevity [9]. A similarly reduced longevity is observed for the *RDL* dieldrin resistance marker in some strains of the malaria vectors *An. gambiae* and *Anopheles stephensi* resistant [7]. Increased longevity in mosquitoes with 119F resistant allele could be associated with the implication of *GSTe2* in oxidative stress. Previously, the *GST* resistance mechanism was shown to protect tissues from oxidative damage in plant hoppers and increase longevity in fruit flies [46] which can be the case for *GSTe2* in this study. In fact, an enhanced ability of the insecticide resistant insects to tolerate oxidative stress has also been implied by the protective role of glutathione *S*-transferases [47]. A study of the *GST* expression in the Hessian fly also found that some *GSTs* could provide protection against toxic oxygen species generated endogenously during development [48]. This could be an explanation of the increased longevity in resistant mosquitoes due to *GSTe2* noticed in this study. This increased longevity of resistant 119F-GTe2 mosquitoes could lead to an increase in malaria transmission in areas where this gene is over-expressed and with a high frequency of L119F-GSTe2 mutation. This could explain recent results showing an elevated *Plasmodium* infection in *An. funestus* populations with the high frequency of the 119F-GSTe2 allele such as in Benin [49], Cameroon [28,50] and Congo [51]. The increased longevity of resistant mosquitoes shows that insecticide could negatively impact the effectiveness of vector control and increase malaria transmission as recently shown for bed nets in Tanzania [52].

### 4.2. Association between L119F Resistance Marker and Larvae/Pupae Formation

Developmental time of the larvae is another key aspect of fitness cost in mosquito populations [53]. This is a very important aspect since, in the presence of natural predators or parasites, any delay in development has the potential to reduce the survival rate of larvae [54]. Here, we found a heterozygote advantage in term of developmental time compared to homozygote resistant and susceptible mosquitoes for the L119F-GSTe2 marker. Such heterozygote advantage was observed also in *kdr* target site resistance although for mating competitiveness in *An. gambiae* from Burkina-Faso [5]. Homozygous susceptible mosquitoes developed faster than homozygous-resistant showing a possible fitness cost of L119-GSTe2 on the larval developmental time. It is possible that despite the fact that all three genotypes were reared in the same container, larvae with 119F resistant allele were less skilled to compete with those with L119 susceptible allele for food and space and the latter ended up developing faster. It is possible, as observed previously in a carboxylesterase-mediated metabolic resistant *Culex pipiens* [55], that over-expression of *GSTe2* is associated with a decreased locomotive performance limiting the ability of resistant mosquitoes to move to feed. This could explain the longer developmental time of resistant RR compared to susceptible SS mosquitoes. Brito et al. observed also that the resistant strain Rock-kdr took more time to develop when competing with the susceptible strain Rock [8]. Such fitness suggests that resistance management strategies such as insecticide rotation could help reverse the resistance if implemented early as homozygote resistant mosquitoes are more likely to be outcompeted by susceptible homozygous but also more exposed to predators in natural breeding sites.

### 4.3. Association between L119F Resistance Marker and Fecundity/Fertility

In this study, the fecundity rate was reduced for 119F-RR homozygous resistant mosquitoes compared to the heterozygotes and the susceptible genotype but no difference in the number of larvae per female was observed. This lower fecundity rate highlights another fitness cost associated with the 119F insecticide resistance allele. Reduction of the ability of resistant mosquitoes to lay eggs was also noted in *Ae. aegypti* [9]. In the same line, Brito et al. noticed that although a load of ingested blood did not differ between Rock and Rock-kdr *Ae. aegypti* females, and the latter displayed a reduction in the rate of insemination and the number of eggs laid [8]. Several factors could contribute to the lower fecundity in resistant females: either a lower fecundity rate or, a lower blood meal size of resistant mosquitoes. Alternatively, it could be due to decreased egg-laying ability. In nature, resistant females would be at a great fitness disadvantage if they spend less time searching for hosts or good oviposition sites or if they are less responsive to predators. In this study, lesser ability of resistant mosquitoes to lay eggs could be linked to lower insemination rates as observed for dieldrin resistant *An. gambiae* and *An. stephensi* females [7]. There are numerous reports describing the reduction in the number of eggs laid by insecticide resistant strains derived from controlled selection experiments. This was the case of fenvalerate resistant *Spodoptera exigua* [56], deltamethrin and diflubenzuron resistant *Cydia pomonella* [57]. In this study, reduced larval hatching rate was not observed as reported by Mebrahtu et al. [58] in *Ae. aegypti* derived from permethrin-resistant specimens. As observed for the development of larvae, a fitness cost observed in term of fecundity could improve the success of potential resistance management strategies particularly if implemented before the frequency of L119F-GSTe2 allele become too high or fixed in a population.

## 5. Conclusions

This study has shown that L119F-GSTe2 mediated metabolic resistance to pyrethroids/DDT likely is associated with negative effects on some life-traits of *An. funestus* field mosquitoes. This supports the assumption that insecticide resistance is associated with a fitness cost showing that resistance management strategies such as insecticide rotation could help reverse resistance if implemented early. However, the increased longevity observed in resistant mosquitoes represents a serious threat for disease control, as increased longevity of 119F/F resistant mosquitoes could lead to an increased level of malaria transmission in areas where this resistance mechanism is predominant as suggested by recent studies.

## Figures and Tables

**Figure 1 genes-09-00645-f001:**
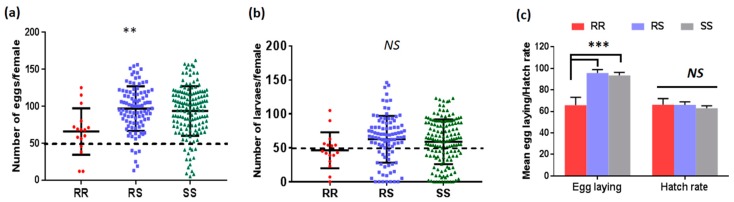
Fecundity and fertility of females with different genotypes at the L119F locus of the *GSTe2* gene: (**A**) Comparison of the number of eggs laid by field-collected female *Anopheles funestus* between the L119F-RR, L119F-RS and L119F-SS genotypes; (**B**) number of larvae produced by females from each genotype; (**C**) hatching rate between the three genotypes. Each dot represents a single egg-laying female. Median value with interquartile range is shown for each distribution. Dotted line indicates females for which at least 50 eggs or larvae were obtained. ** Difference between genotypes was significant (*p* < 0.01) in term of eggs laying by Kruskal–Wallis non-parametric test whereas the number of larvae produced and the hatch rate did not differed significantly. ***: significant difference at *p* < 0.001; NS: not significant.

**Figure 2 genes-09-00645-f002:**
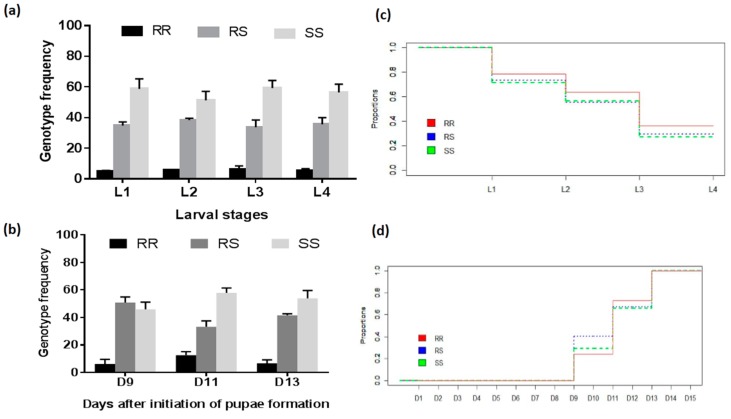
Distribution of the L119F-GSTe2 genotypes at different time-points of the development of immature stages. (**a**) Histogram of the variation in genotypes frequency during the development of larvae (L1, L2, L3, and L4 represent different larval stages) and pupae formation (**b**); (**c**) the proportion of larvae surviving at each developmental stage from hatching (D1) to formation of the pupae; (**d**) the proportion of pupae obtained in D9, D11 and D13 of development. Colored bars and lines indicate respectively 119F/F-RR, L119F-RS and L/L119-SS genotypes. Standard error (*n* = 3) are also indicated for the histograms.

**Figure 3 genes-09-00645-f003:**
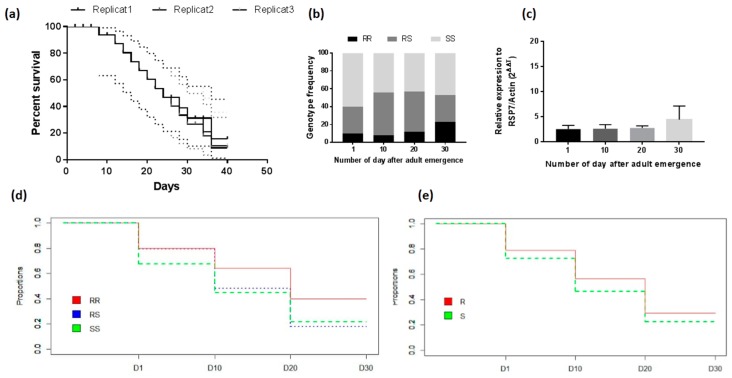
Influence of L119F-GSTe2 on the adult longevity of *An. funestus*. (**a**) Survival curve F1of adults from natural populations and maintained under laboratory conditions: Mean percentage of mortality and 95% confidence interval (CI) were presented; (**b**) distribution of L119F-GSTe2 genotypes at different time in the survived mosquitoes; (**c**) differential expression by quantitative reverse-transcription polymerase chain reaction of *GSTe2* genes in alive mosquitoes at different time points compared with the susceptible lab strain FANG. Error bars represent standard error of the mean; (**d**,**e**) Variation in the proportion of adults surviving at the different time points after the emergence into adult according to the L119F genotypes and alleles respectively.

**Figure 4 genes-09-00645-f004:**
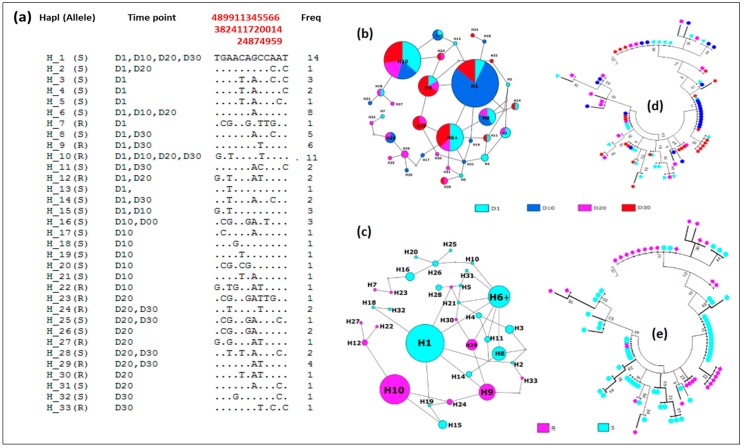
Genetic diversity parameters of *GSTe2* in *An. funestus* s.s. from Mibellon (Cameroon) in relation to the longevity of adult mosquitoes. (**a**–**c**) haplotype diversity in relation to the alleles at different time points; (**d**,**e**) phylogenetic trees (using a maximum likelihood method) between mosquitoes at the different time point after the emergence of F1 adult with respect to the alleles.

**Table 1 genes-09-00645-t001:** Assessment of the association between L119F-GSTe2 genotypes and the ability of females to lay eggs. SS: homozygote susceptible; RR: homozygote resistant; RS: heterozygote; * significant difference *p* < 0.05.

Genotypes	L119F-GSTe2 and Oviposition	
	Odds Ratio	*p*-Value
SS vs. RR	2.93 (1.66–5.18)	0.0001 *
SS vs. RS	2.06 (1.45–2.92)	0.000001 *
RS vs. RR	2.68 (1.51–4.77)	0.0002 *

**Table 2 genes-09-00645-t002:** Association between L119F-GSTe2 genotypes and pupae formation. * significant difference

Genotypes	Pupae D9 vs. Pupae D11		Pupae D11 vs. Pupae D11	
	Odds Ratio	*p*-Value	Odds Ratio	*p*-Value
RS vs. RR	5.26 (2.24–12.34)	<0.0001 *	1.04 (0.73–1.49)	0.42
RS vs. SS	1.39 (0.89–2.17)	0.08	1.38 (0.98–1.87)	0.03 *
SS vs. RR	9.66 (4.17–22.40)	<0.0001*	1.40 (1.01–1.95)	0.02 *

**Table 3 genes-09-00645-t003:** Association between L119F-GSTe2 genotypes and adult longevity.

Genotypes	D_1 x_ D_10_		D_10 x_ D_20_		D_20 x_ D_30_	
	Odds Ratio	*p*	Odds Ratio	*p*	Odds Ratio	*p*
RR vs. RS	3.75 (1.21–11.29)	0.019	3.83 (1.56–9.41)	0.0023	2.2 (1.04–4.64)	0.050 ^S^
RR vs. SS	7.5 (2.64–21.28)	0.000006	3.83 (1.56–9.41)	0.0059	2.1 (0.98–4.45)	0.13
RS vs. SS	1.30 (0.75–2.24)	0.41	1.04 (0.57–1.90)	1	1.61 (0.80–3.22)	0.22

**Table 4 genes-09-00645-t004:** Genetic diversity parameters of *GSTe2* sequences according to the age of mosquitoes and the L119F genotypes.

	2*n*	S	h	hd	π	D	D*
D1	24	11	15	0.95	0.005	−0.11 ns	−0.96 ns
D10	24	10	11	0.79	0.003	−0.44 ns	0.97 ns
D20	16	11	14	0.98	0.006	0.21 ns	0.41 ns
D30	24	9	12	0.92	0.003	0.02 ns	0.28 ns
TOTAL	88	12	33	0.94	0.004	0.44 ns	1.55 ns

2*n*, number of sequences; D, Tajima’s statistics; D, D* Fu and Li’s statistics; h, number of haplotypes; hd, haplotype diversity; ns, not significant; π, nucleotide diversity; S, number of polymorphic sites.

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
