# Peer review of "Fitness Costs of the Glutathione S-Transferase Epsilon 2 (L119F-GSTe2) Mediated Metabolic Resistance to Insecticides in the Major African Malaria Vector Anopheles Funestus"

_genes, 2018, doi:10.3390/genes9120645_

Round 1

Reviewer 1 Report

This is a technically well executed study and good manuscript, prepared by experts in the field, that worth to be published in Genes. I only have a comment, which would request the authors to consider throughout their manuscript:

Despite very careful comprehensive and detail experimental, I dont think there is enough  evidence and justification at this stage (i.e. without further work to support any possible functional link) to claim "Impact of L119-GSTe2 on life traits", anywhere in the manuscript (for example lines 28 - a fitness cost of L119 mutation on .., 110, 111, 141, 178, 235 - fitness cost "imposed by", 286..). The term "Association between L119-GSTe2 genotypes and .." that has been used in the Table titles is far more appropriate for this study. 

Author Response

We thanks the reviewer for this remark. We agree that the impact of L119F-GSTe2 mutation that we observed in this study can be due to pleiotropic effects in this gene itself or a consequence of closely linked genes.  As the author suggested, because we did not conduct a study to support any possible functional link, the term "association", is more appropriate than "Impact". We have revised the entire manuscript to correct statements like those mentioned by the reviewer to avoid speculative sentences which could tend to overstate the outcomes of our findings.  

Reviewer 2 Report

Dear Editor,

I have carefully read the submission entitled ‘Fitness cost of the glutathione S-transferase epsilon 2 (L119F-GSTe2) mediated metabolic resistance to insecticides on the major African malaria vector Anopheles funestus’, by Tchouakui and colleagues.

The authors try to establish a correlation between a specific insecticide resistance mutation and a possible fitness cost for the mosquito vector that harbors this mutation. To address this, they brought females from the field with different genotypes in respect to the mutation status (RR, RS, and SS), let them oviposit in the laboratory and measured different quality control parameters including fecundity, fertility, immature development duration, and adult longevity.

The topic is interesting and a substantial amount of work has been performed. However, in my opinion, the overall set up of the study is not appropriate for the questions posed, leading to my major concern: experiments leading to results that do not answer the specific questions and can be ‘translated’ in various ways. Therefore, my suggestion is rejection. Below you can find my major and minor concerns, hoping at least that they will be useful to the authors.

Major concern:

1.       Experimental set up, results, discussion, and main conclusions: in my opinion the correlation presented between the three genotypes and the fitness cost is not justified. I am not saying that it is not real, I am saying that the specific experiments do not lead to this correlation. To further clarify this: the authors brought in the lab several females and made different measurements such as fertility, fecundity, immature development, and longevity. After that, females were genotyped in respect to one specific gene that is related with insecticide resistance. The problem is that females carrying different genomic backgrounds have been ‘clustered’ in three main groups based on the RR-RS-SS phenotype. Taking into account the importance of the genetic background for the behavior of the vector, I am not convinced that the interpretation of the results is so straightforward. An approach where the three resistance related genotypes would be introduced in a common genetic background would limit (even eliminate) the effect of natural polymorphism(s). To make it simpler, introducing the mutation(s) through an introgression experiment in a laboratory population and having the three combinations in this background would make the analysis much easier and ‘convincing’. The fitness cost (or advantage) described could be the accumulated result of numerous other mutations and loci that have been either selected or randomly associated with the gene under study. However, this is my perspective and the editor may want to have the opinion of another reviewer as well.

2.       Additional concerns:

a.       Substantial editing is needed: there are many grammar and language mistakes throughout the submission. Extensive editing is needed.

b.       Referencing and reference list: there are problems both in the within the manuscript citations and with the reference list.

c.       Materials and methods: the statistical analysis used to assess differences in different measurements is not described properly.

Author Response

Reviewer 2: Comments and Suggestions for Authors

Dear Editor,

I have carefully read the submission entitled ‘Fitness cost of the glutathione S-transferase epsilon 2 (L119F-GSTe2) mediated metabolic resistance to insecticides on the major African malaria vector Anopheles funestus’, by Tchouakui and colleagues.

The authors try to establish a correlation between a specific insecticide resistance mutation and a possible fitness cost for the mosquito vector that harbors this mutation. To address this, they brought females from the field with different genotypes in respect to the mutation status (RR, RS, and SS), let them oviposit in the laboratory and measured different quality control parameters including fecundity, fertility, immature development duration, and adult longevity.

The topic is interesting and a substantial amount of work has been performed. However, in my opinion, the overall set up of the study is not appropriate for the questions posed, leading to my major concern: experiments leading to results that do not answer the specific questions and can be ‘translated’ in various ways. Therefore, my suggestion is rejection. Below you can find my major and minor concerns, hoping at least that they will be useful to the authors.

Major concern:

1.           Experimental set up, results, discussion, and main conclusions: in my opinion the correlation presented between the three genotypes and the fitness cost is not justified. I am not saying that it is not real, I am saying that the specific experiments do not lead to this correlation. To further clarify this: the authors brought in the lab several females and made different measurements such as fertility, fecundity, immature development, and longevity. After that, females were genotyped in respect to one specific gene that is related with insecticide resistance. The problem is that females carrying different genomic backgrounds have been ‘clustered’ in three main groups based on the RR-RS-SS phenotype. Taking into account the importance of the genetic background for the behavior of the vector, I am not convinced that the interpretation of the results is so straightforward. An approach where the three resistance related genotypes would be introduced in a common genetic background would limit (even eliminate) the effect of natural polymorphism(s). To make it simpler, introducing the mutation(s) through an introgression experiment in a laboratory population and having the three combinations in this background would make the analysis much easier and ‘convincing’. The fitness cost (or advantage) described could be the accumulated result of numerous other mutations and loci that have been either selected or randomly associated with the gene under study. However, this is my perspective and the editor may want to have the opinion of another reviewer as well.

We thanks the reviewer for this comment. We agree that fitness cost of insecticide resistance in the absence of selection in the life history traits of mosquito’s population can be achieved as a result of pleiotropic effects in the resistance genes themselves or as a consequence of a hitchhiking effect. For this reason when mosquitoes are originated from different locations, an introgression experiment is needed in the laboratory to bring those population to the same genetic background.  But in our study, all mosquitoes that we used were collected in the same location at the same period. So they have a similar genetic background and the main difference between those mosquitoes was the mutation at the L119F locus of the GSTe2 gene. For this reason, we expect that the main differences observed in the mosquitoes’ life traits in the absence of selection pressure are more likely linked with this mutation, although we cannot exclude possible effects associated with very closely linked genes. This has been added in discussion. Overall, the experimental design we followed is the best to address this question as the only difference between the F1 mosquitoes or even F0 is their genotypes for the L119F-GSTe2 markers. The other advantage of using such field collected mosquitoes than introgressed strains is that we have a direct information on the influence of this marker in natural populations harboring it.

2.       Additional concerns:

a.       Substantial editing is needed: there are many grammar and language mistakes throughout the submission. Extensive editing is needed.

We have revised the whole manuscript extensively for English grammar and language mistakes and we believe that changes made have significantly improved the quality of English.

b.       Referencing and reference list: there are problems both in the within the manuscript citations and with the reference list.

We have double-checked the references in the text and reference list and made changes when needed.

c.       Materials and methods: the statistical analysis used to assess differences in different measurements is not described properly.

More explanation is given on the statistical analysis used to assess differences in different measurements as you can see on the track changes version.

Round 2

Reviewer 2 Report

Dear Editor,

I carefully read the revised version of the submission entitled ‘Fitness costs of the glutathione S-transferase epsilon 2 (L119F-GSTe2) mediated metabolic resistance to insecticides in the major African malaria vector Anopheles funestus’ by TCHOUAKUI and colleagues. I am sorry to say that, in my opinion, the authors did not ‘do their best’ effort to provide a really improved version to address the concerns raised. My suggestion is major revision, to allow the authors present both their arguments and the limitations of their study in the new version. However, I have to say again that the editor is welcome to ask the opinion of another reviewer if he does not agree with my concerns (there can sometimes be differences in the perspective or interpretation of results).

Below you can find my major concerns.

1.       Title, main question, experiments, results, and discussion:

a.       although the authors agree that the causal relationship between the mutation and different fitness parameters is not clearly documented, they do not really address it to the extend they should. Although they changed different section headings (from ‘impact’ to ‘association’, for example), there are no real changes in the results and discussion section to reflect this. I must remind here what I stated before: I do not say that there is no causal relationship, I say that this relationship is not documented from the experiments performed.

b.       The authors say in their response that the natural population used is expected to have similar genetic background. Although this may be true, it is far from true that there will be no polymorphisms that could affect rather complex traits, like those involved in different fitness parameters. As an additional concern, the authors also say that the plasmodium infection is varying in this natural population and although such a parasite may interfere with fitness parameters, the authors do not take this into account, suggesting that the population is rather ‘homogenous’. Since the authors used the specific approach, they should at least clearly specify the limitations of their study, such as the incomplete knowledge on the genetic diversity and vectorial capacity/status of a) the natural population they used and b) the females used for their study.

c.       In the revised version some of the editing is in the direction of a documented causal relationship between the mutation and fitness parameters, although what was asked was the opposite. See for example lines 21-22, 28, 563-564.

2.       English language and editing performed: although the authors tried to perform some substantial English editing, the outcome is not the expected. In fact, some of the new parts need also English language editing. See for example lines 67-69 (confusing), 78-79 (‘a mechanism that have been…’), 170 (‘lowest’) and other.

Author Response

Response to Reviewer 2 Comments

Point 1 Title, main question, experiments, results, and discussion:

a.   Although the authors agree that the causal relationship between the mutation and different fitness parameters is not clearly documented, they do not really address it to the extend they should. Although they changed different section headings (from ‘impact’ to ‘association’, for example), there are no real changes in the results and discussion section to reflect this. I must remind here what I stated before: I do not say that there is no causal relationship, I say that this relationship is not documented from the experiments performed.

Response: We need to clarify that in the past, we have clearly established that the L119F-GSTe2 mutation is the key factor conferring resistance to both pyrethroids and DDT in this mosquito species in West and Central Africa. This extensive work was published in 2014 in Genome Biology (Riveron et al 2014). There is a very strong evidence that there is a causal relationship between this mutation and the fitness cost we assessed because the L119F-GSTe2 has been shown to be the only mutation in this gene inducing a phenotype variation. It is also a well-established procedure when studying fitness cost to correlate genotypes of a resistance marker (kdr, Ace-1, rdl etc) with various physiological traits as we have done in this study. Previous publications could be consulted for this matter (Britto et al PLoS ONE 2013; Platt et al 2015 Heredity etc). By already changing from “impact” to “association” is by itself an indication that we took into account the recommendation of the reviewer on the possibility of potentially linked factors (although from transcriptomic, whole genome sequencing and fine-scale mapping we have established that L119F is the only factor here). However, making that change of emphasis should not impact the observation that we made of a significant correlation between genotypes of L119F- and fitness cost in the results or discussion. We believe that because we had previously extensively established that the L119F is the causative mutation for pyrethroid/DDT resistance in this genomic region, we did not need to repeat such study again before starting the fitness cost. This is a logical approach as we are building on previous work. Therefore, if we consider that our goal was to establish the influence of the L119F on the fitness of An. funestus, we think that the approach taken is sound and addresses this question.

b.       The authors say in their response that the natural population used is expected to have similar genetic background. Although this may be true, it is far from true that there will be no polymorphisms that could affect rather complex traits, like those involved in different fitness parameters. As an additional concern, the authors also say that the plasmodium infection is varying in this natural population and although such a parasite may interfere with fitness parameters, the authors do not take this into account, suggesting that the population is rather ‘homogenous’. Since the authors used the specific approach, they should at least clearly specify the limitations of their study, such as the incomplete knowledge on the genetic diversity and vectorial capacity/status of a) the natural population they used and b) the females used for their study.

Thanks to the reviewer for the comment. In fact the advantage of using field collected mosquitoes is that we have a direct information on the influence of insecticide resistance on the natural populations harbouring it. For the population of mosquitoes we used for this study, before assessing the fitness cost on other life traits, we did a thorough investigation on the impact of the L119F mutation on Plasmodium infection and we found no significant difference between the three genotypes in term of infection (paper submitted) so we can confirm that Plasmodium infection had no influence on the differences observed in the fecundity as this is the only fitness parameter based on field collected female (F0) mosquitoes which are the only one that could have been infected. Other fitness parameters we assessed are based on F1 mosquitoes reared in lab condition so there is no interference from infection status as they are all non-infected.  We extensively sequenced the genomic regions spanning GSTe2 genes and only the L119F mutation is associated with insecticide resistance and all other substitutions were segregating irrespective of the resistance phenotype (Riveron et al 2014 Genome Biology; Barnes et al 2017 PLoS Genetics). That is why we can confidently state that the L119F is the key mutation distinguishing the different mosquitoes assessed and thus the association found is linked with the fitness cost of the resistance as it is the only variation here between these mosquitoes.

c.     In the revised version some of the editing is in the direction of a documented causal relationship between the mutation and fitness parameters, although what was asked was the opposite. See for example lines 21-22, 28, 563-564.

As we explained above, the L119F is clearly the key mutation conferring resistance and the only genetic variant identified in the genomic region which is associated with pyrethroid/DDT resistance. This is a fact that we have previously extensively demonstrated using a combination of approaches from phenotype assays, transcriptomic (with GSTe2 gene as the main over-expressed gene), allelic variant (detecting the 119F allele only in resistant mosquitoes), X-ray crystallography revealing that the 119F allele enlarges the substrate binding pocket to better metabolise the insecticides. It also included functional analyses in vivo and in vitro showing that 119F allele is a better metabolizer of insecticides than L119 and finally the gene and surrounding regions were extensively sequenced Africa-wide showing that the unique variant linked with resistance is the L119F. All these evidences already published (Riveron et al 2014) and further validated in the field later (Djouaka et al 2016; Menze et al 2016; 2018) support the causal relationship between L119F and resistance and thus associated fitness cost. However, to avoid to engage in a protracted argument on this issue we did nevertheless accept to include the possibility of potential linked genetic factors as suggested by the reviewer. However, we feel that based on the weight of existing evidences of the role of L119F, that the data presented have been discussed in a balanced way between a definitive causal relationship (our opinion) and the potential presence of other factors (as suggested by the reviewer). 

Point 2.  English language and editing performed: although the authors tried to perform some substantial English editing, the outcome is not the expected. In fact, some of the new parts need also English language editing. See for example lines 67-69 (confusing), 78-79 (‘a mechanism that have been…’), 170 (‘lowest’) and other.

We have double-checked the entire manuscript for some English mistakes. Thanks to the reviewer

Round 3

Reviewer 2 Report

Dear Editor,

the authors have done an effort to provide an improved version of the manuscript. Some of my initial reservations have not been totally addressed, but, as I stated in the previous round of revision, sometimes is also a matter of 'perception'.

I have no further comments on this manuscript.